# ADAM ACCUMULATION TO REDUCE MEMORY FOOT-PRINTS OF BOTH ACTIVATIONS AND GRADIENTS FOR LARGE-SCALE DNN TRAINING

## ABSTRACT

Running out of GPU memory has become a main bottleneck for large-scale DNN training. How to reduce the memory footprint during training has received intensive research attention. We find that previous gradient accumulation reduces activation memory but fails to be compatible with gradient memory reduction due to a contradiction between preserving gradients and releasing gradients. To address this issue, we propose a novel optimizer accumulation method for Adam, named Adam Accumulation (AdamA), which enables reducing both activation and gradient memory. Specifically, AdamA directly integrates gradients into optimizer states and accumulates optimizer states over micro-batches, so that gradients can be released immediately after use. We mathematically and experimentally demonstrate AdamA yields the same convergence properties as Adam. Evaluated on transformer-based models, AdamA achieves up to 23% memory reduction compared to gradient accumulation with less than 2% degradation in training throughput. Notably, AdamA can work together with memory reduction methods for optimizer states to fit $1.26\times$~$3.14\times$ larger models over PyTorch and DeepSpeed baseline on GPUs with different memory capacities.

## 1 INTRODUCTION

The past few years have witnessed the remarkable achievements of large-scale DNN models across domains from computer vision to natural language processing (Devlin et al., 2018; Radford et al., 2019; Dosovitskiy et al., 2020; Smith et al., 2022). Training such big models requires massive powerful GPUs with indispensable large memory capacity, which is prohibitively expensive and inaccessible to most researchers. Even for fine-tuning a large pre-trained model where computational power is a less critical factor, **running out of memory** is increasingly becoming the first and foremost serious limitation (Ren et al., 2021; Rajbhandari et al., 2021).

Recently, there has been an explosion of interest around methods to reduce the memory footprint during model training (Sohoni et al., 2019; Rajbhandari et al., 2020; Pudipeddi et al., 2020; Chen et al., 2016; Shazeer & Stern, 2018). However, there is hardly a one-size-fits-all solution to address the out-of-memory issue for two reasons. Firstly, many memory reduction methods usually come at the cost of sacrificing convergence (Mostafa & Wang, 2019; Micikevicius et al., 2017) or training throughput (Chen et al., 2016; Pudipeddi et al., 2020). It remains unclear how significant the cost of one method or a combination of methods would be for different models before testing. Secondly, the ratio of the memory footprint of various parts (e.g., weights, gradients, optimizer states, activations) varies with the model and training configurations. No single method always performs best in different cases.

Among memory reduction methods, *gradient accumulation* and *gradient release* are two effective methods to reduce activation memory and gradient memory, respectively (Huang et al., 2019; Pudipeddi et al., 2020). Both methods have no negative impact on model convergence and training throughput. Unfortunately, these two methods are inherently mutually exclusive. Gradient accumulation reduces the *activation memory* by splitting a mini-batch into a sequence of micro-batches and accumulating the gradients of all micro-batches. Gradient release reduces the *gradient memory* by freeing up the gradient-occupied space in a layer-by-layer manner. The contradiction preventing

the two from being used together is one must preserve accumulated value of gradients until the last micro-batch, but the other releases the gradients immediately after use. Saving activations or gradients, previous works prefer the former as activations usually consume the most memory during training, while the gradients memory can be ignored when models are small. However, with the ever-increasing model size, the gradient memory consumption cannot be ignored.

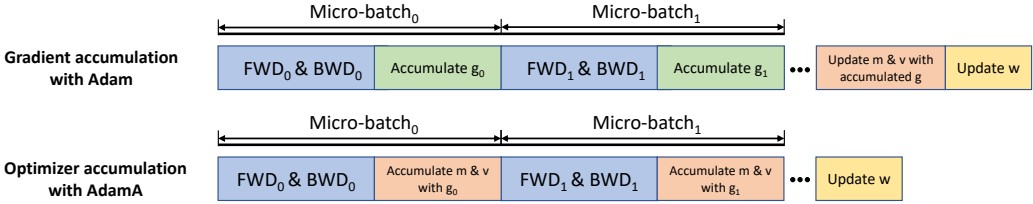

Figure 1: AdamA to integrate gradients into optimizer states and accumulate optimizer states over micro-batches.

To enable saving memory footprints of both activations and gradients, we propose a novel optimizer accumulation method for large-scale DNN training with Adam, named **Adam Accumulation (AdamA)**, which can still maintain the convergence and training throughput. Specifically, instead of accumulating gradients, AdamA integrates gradients into optimizer states ($m$ and $v$ in Adam) immediately after the gradients are produced, and accumulates optimizer states sequentially over micro-batches, as shown in Figure 1. This subtle change of directly integrating gradients to optimizer states makes the memory space for whole model gradients no longer needed, eliminating the aforementioned contradiction between preserving gradients and releasing gradients. Consequently, AdamA can reduce the gradient memory to $1/M$ of the original ($M$ is the number of layers), and the activation memory to $1/N$ of the original ($N$ is the number of micro-batches). We further mathematically and experimentally demonstrate AdamA performs the same as standard Adam in terms of the convergence properties and final model accuracy, although the optimizer update of AdamA deviates a little from standard Adam. Notably, AdamA is complementary to previous methods that reduce weights and optimizer states, providing a possibility to achieve an even higher memory reduction rate.

We evaluate AdamA on both language and vision tasks, with the typical transformer architecture and convolution architecture. Our experimental results show that AdamA performs the same convergence properties as Adam. Compared with gradient accumulation baseline, AdamA can reduce memory footprint up to 23% with less than 2% degradation in training throughput. We further combine AdamA with DeepSpeed ZeRO-DP $P_{os}$, which aims to reduce optimizer states in distributed data parallel scenario. Training with AdamA, a DGX system can fit a model **1.26×~3.14×** larger over PyTorch and DeepSpeed baseline can do.

Our contributions can be summarized as follows:

- We propose AdamA, a novel optimizer accumulation method to enable reducing memory footprints of activations and gradients simultaneously. Compared with gradient accumulation baseline, AdamA can save up to 23% memory footprint.

- We conduct a convergence analysis for AdamA. Mathematical and experimental results on real workloads show AdamA performs the same convergence properties as Adam.

- We implement the training pipeline of AdamA with Pytorch and DeepSpeed. The system is easy to use and incurs less than 2% effect on training throughput.

## 2 BACKGROUND AND RELATED WORK

The memory footprint during model training can be categorized into four parts: weights, gradients, optimizer states and activations. As different models, optimizers, or batch sizes lead to different ratios of the four parts, many works have been proposed to reduce them accordingly.

**Reducing weight and optimizer state memory.** In model training iterations, weights and optimizer states inherently have the *temporal dependency*, i.e., the values at time step $t$ update on the basis of

their values at time step $t-1$. Hence, the training system must maintain weights and optimizer states for updates between two consecutive iterations. To reduce the weight and optimizer state memory, many *compression-based* methods (e.g., sparsification, quantization and matrix approximation) have been proposed, but often sacrifice the convergence or end model accuracy (Mostafa & Wang, 2019; Micikevicius et al., 2017; Shazeer & Stern, 2018).

**Reducing activation and gradient memory.** Activations and gradients are computed and used only inside each training iteration, indicating a potential to release the memory occupation after finished. *Gradient accumulation* and *gradient release* are effective methods to reduce activations and gradients, respectively (Huang et al., 2019; Pudipeddi et al., 2020). The key idea behind *gradient accumulation* is to split a mini-batch into several micro-batches. This method computes the gradients of micro-batches sequentially and accumulates them to reduce the memory footprint of activations as well as to keep the same convergence properties as the original mini-batch. *Gradient release* executes the backward process in a layer-by-layer manner, which immediately releases the gradient-occupied memory after the weight updating is finished, so that the memory allocated for gradients can be reduced from the size of the whole model size to the size of the maximum layer.

**The contradiction between gradient accumulation and gradient release.** Unfortunately, gradient accumulation to save activation memory and gradient release to save gradient memory are mutually exclusive. Because one must maintain all gradients for accumulation until the last micro-batch, while the other frees up the gradients immediately after use. Our proposed AdamA resolves this contradictory and enables saving both activation and gradient memory. Please note that our method is complementary to previous memory reduction methods (e.g., checkpointing (Chen et al., 2016), Adafactor (Pudipeddi et al., 2020), offloading (Rajbhandari et al., 2021; Ren et al., 2021; Pudipeddi et al., 2020)), and can be applied together with these methods to achieve even higher memory reduction rate.

## 3 METHODS

### 3.1 ADAM ACCUMULATION (ADAMA)

As mentioned, gradient accumulation to save activation memory contradicts gradient release to save gradient memory. The core reason is that gradient accumulation accumulates gradients till the last micro-batch, so that the gradient memory of the whole model must be preserved. Intuitively, as gradients are eventually used to update the optimizer states ($m$ and $v$ in Adam), if we can integrate gradients into optimizer states in advance, the gradients memory can be released, thus resolving this dilemma. Inspired by this insight, we for the first time propose an optimizer accumulation method, namely AdamA, that integrates gradients into optimizer states immediately after produced and then accumulates optimizer states sequentially over micro-batches.

---

**Algorithm 1** Adam v.s. AdamA with micro-batches

---

**Initialize** $\theta_0$, $m_0 \leftarrow 0$ , $v_0 \leftarrow 0$, $t \leftarrow 0$, $N \leftarrow \# \ of \ microbatches$
**while** $\theta_t$ *not converged* **do**
  $t \leftarrow t + 1$
  **for** *each micro-batch $i$ in a mini-batch* **do**
    $g_{t,i} \leftarrow \frac{1}{N}\nabla_\theta f_{t,i}(\theta_{t-1})$
  **end**
  $m_t = \beta_1 m_{t-1} + (1-\beta_1)\sum_{i=0}^{N-1} g_{t,i}$
  $v_t = \beta_2 v_{t-1} + (1-\beta_2)[(\sum_{i=0}^{N-1} g_{t,i})^2 v.s. \sum_{i=0}^{N-1}(g_{t,i}^2)]$
  **Update**
  $\hat{m}_t \leftarrow \frac{m_t}{1-\beta_1^t}, \hat{v}_t \leftarrow \frac{v_t}{1-\beta_2^t}, \theta_t \leftarrow \theta_{t-1} - \frac{\alpha \hat{m}_t}{\sqrt{\hat{v}_t}+\epsilon}$

**end**

---

Algorithm 1 illustrates the difference between standard Adam and our proposed AdamA, in the case of micro-batching. Standard Adam first accumulates gradients of all micro-batches, then updates $m_t$ with the accumulated gradients and $v_t$ with the square of the accumulated gradients (as shown in

the blue text). Different from the $v_t$ update mechanism in Adam, our proposed AdamA updates $v_t$ through accumulating the square of gradients generated from each micro-batch. This slight change in AdamA allows that gradients can be used and released immediately once they are generated, leading to a significant reduction on gradient memory during training. In order to analyze AdamA's impact on model convergence, we mathematically prove that AdamA yields the same convergence rate as Adam (shown in Section 3.2), and experimentally demonstrate AdamA performs the same as Adam in vision and language tasks (shown in Section 4.1).

In Algorithm 2, we show the detailed training pipeline of AdamA to reduce both activation and gradient memory. Similar to gradient accumulation, AdamA divides a mini-batch of training data into several micro-batches to reduce activation memory to $\frac{1}{N}$ of the original without micro-batches. During the backward pass of each micro-batch, once the gradients of a layer ($g_{t,i,j}$) are produced, $g_{t,i,j}$ and $g_{t,i,j}^2$ will be accumulated to the optimizer states of this layer ($m_{t,j}$ and $v_{t,j}$), respectively. In this process, $g_{t,i,j}$ memory is released after the accumulation procedure. As a result, the peak memory allocated for gradients can be reduced to only $\frac{1}{M}$ of the full model gradient size.

---

**Algorithm 2** The training pipeline using AdamA to reduce both activation and gradient memory

---

**Initialize** $\theta_0, m_0 \leftarrow 0, v_0 \leftarrow 0, t \leftarrow 0, N \leftarrow \# \ of \ micro - batches, M \leftarrow \# \ of \ layers$
**while** $\theta_t$ *not converged* **do**
  $\quad t \leftarrow t + 1, m_t \leftarrow \beta_1 m_{t-1}, v_t \leftarrow \beta_2 v_{t-1}$
  $\quad$**for** *each micro-batch **i** in a mini-batch* **do**
    $\quad\quad$ // Reduce activation memory to 1/N of the original without micro-batches
    $\quad\quad$**for** *each layer **j** in backward computing* **do**
      $\quad\quad\quad g_{t,i,j} \leftarrow \frac{1}{N} \nabla_\theta f_{t,i,j}(\theta_{t-1})$
      $\quad\quad\quad$ Assign memory for $g_{t,i,j}$  // Reduce gradient memory to 1/M of full model gradients
      $\quad\quad\quad m_{t,j} \leftarrow m_{t,j} + (1 - \beta_1)g_{t,i,j}$
      $\quad\quad\quad v_{t,j} \leftarrow v_{t,j} + (1 - \beta_2)g_{t,i,j}^2$
      $\quad\quad\quad$ Release memory for $g_{t,i,j}$  // The $g_{t,i,j}$ values are not needed any more
    $\quad\quad$**end**
  $\quad$**end**
  $\quad$**Update**
  $\quad \hat{m}_t \leftarrow \frac{m_t}{1-\beta_1^t}, \hat{v}_t \leftarrow \frac{v_t}{1-\beta_2^t}, \theta_t \leftarrow \theta_{t-1} - \frac{\alpha \hat{m}_t}{\sqrt{\hat{v}_t} + \epsilon}$
**end**

---

### 3.2 CONVERGENCE ANALYSIS

In this section, we demonstrate the convergence properties of AdamA. Adam (Kingma & Ba, 2014) is a optimization method that adaptively rescales the updating vector with second moment of the gradient. Compared with Adam, AdamA has the same updating direction (i.e., $m_t$), but different adaptive scaling length (i.e., $\frac{1}{\sqrt{v}}$). We refer to Adam's proof methods to show that AdamA has the same theoretical convergence properties as Adam.

Following analysis method in the online learning framework(Zinkevich, 2003), we define $f_t$ as the convex cost function at time t, and $\theta_t$ as the parameter we predict. We evaluate the convergence properties of AdamA using the regret $R(T) = \sum_{t=1}^{T}[f_t(\theta_t) - f_t(\theta^*)]$, which is the sum of all the previous difference between our prediction $f_t(\theta_t)$ and the best fixed point parameter $f_t(\theta^*)$ (Kingma & Ba, 2014). In Theorem 1, we guarantee that AdamA has the same regret bound $O(\sqrt{T})$ with Adam and the detailed proof is given in the appendix. We define the vector $g_{1:T,i,\ b} \in \mathbb{R}^t$ as the $i^{th}$ dimension of gradients from the $b^{th}$ micro-batch in one mini-batch till $T$. Following Adam paper, Theorem 1 holds when the learning rate $\alpha_t$ is decaying at a rate of $t^{-\frac{1}{2}}$ and first moment running average coefficient $\beta_{1,t}$ decay exponentially with $\lambda$. (Kingma & Ba, 2014)

**Theorem 1.** Assume $\beta_1, \beta_2 \in [0, 1)$ satisfy $\gamma = \frac{\beta_1^2}{\sqrt{\beta_2}} < 1$. $N$ is the number of micro-batches in a mini-batch. The function $f_t$ has bounded gradients, $\|\nabla f_t(\theta)\| \leq G, \|\nabla f_t(\theta)\|_\infty \leq G_\infty$ for all $\theta \in R^d$. For any $m, n \in \{1, ..., T\}$, the distance between any $\theta_t$ generated by AdamA is bounded,

which can be presented as $\|\theta_n - \theta_m\|_2 \leq D$, $\|\theta_n - \theta_m\|_\infty \leq D_\infty$. AdamA achieves the following guarantee, for all $T \geq 1$.

$$R(T) \leq \frac{D^2}{2\alpha(1 - \beta_1)} \sum_{i=1}^{d} \sqrt{T\hat{v}_{T,i}} + \frac{\alpha(\beta_1 + 1)G_\infty}{(1 - \beta_1)\sqrt{1 - \beta_2}} \sum_{i=1}^{d} \sum_{b=1}^{N} \|g_{1:T,i,\,b}\|_2 + \sum_{i=1}^{d} \frac{D_\infty^2 G_\infty \sqrt{1 - \beta_2}}{2\alpha(1 - \beta_1)(1 - \lambda)^2}$$

In Corollary 1, we show the average regret of AdamA is $O(\frac{1}{\sqrt{T}})$, which is the same as Adam. It is obvious the limit of the average regret is 0 when T gets larger.

**Corollary 1.** Assume that the function $f_t$ has bounded gradients, $\|\nabla f_t(\theta)\| \leq G$, $\|\nabla f_t(\theta)\|_\infty \leq G_\infty$ for all $\theta \in R^d$. For any $m, n \in \{1, ..., T\}$, the distance between any $\theta_t$ generated by AdamA is bounded, which can be presented as $\|\theta_n - \theta_m\|_2 \leq D$, $\|\theta_n - \theta_m\|_\infty \leq D_\infty$. Combining Theorem 1 with the upper bound $\sum_{i=1}^{d} \sum_{b=1}^{N} \|g_{1:T,i,\,b}\| \leq dG_\infty \sqrt{T}$ we can get the average regret of AdamA:

$$\frac{R(T)}{T} = O(\frac{1}{\sqrt{T}})$$

### 3.3 System Implementation to Reduce Memory Footprint

Our system design is intended to reduce memory and incur little impact on training throughput at the same time. The challenge is to achieve efficient memory allocation and low communication cost.

In the single device scenario, the model forward computing is done as normal. When doing backward computing, we interleave back propagation and optimizer states update. For each model layer, gradients are accumulated to the corresponding optimizer states and immediately released. We implement this mechanism in PyTorch with backward hook to insert the optimizer states accumulation and layer gradient release operations. It should be mentioned that frequent memory allocation and free operations are costly. Nevertheless, the deep learning framework would maintain a memory pool for tensor memory assignment and release. This prevents the heavy overhead of system calls.

In the distributed data parallel scenario, the same operations apply except that gradients need to be all-reduced among distributed devices. When training with AdamA, the straightforward implementation is to insert layer gradients all-reduce operation in each micro-batch to update optimizer states. And yet, compared with standard Adam procedure, where all-reduce only needs to be done once after all micro-batches, the communication cost would be increased from O(1) to O(N) in one mini-batch (N is accumulation steps). To reduce the communication volume, we choose to all-reduce optimizer states instead of gradients. In this way, local optimizer states are updated on each device and synchronized at the end of the mini-batch with PyTorch all-reduce API. Therefore, the communication volume stays constant in one mini-batch. The details about our system design can be found in Appendix.

## 4 Experiments

In this section, we evaluate AdamA on both vision and language tasks. We show that AdamA does no harm to the convergence properties compared with Adam. Then, we demonstrate the memory reduction result of AdamA and its impact on training throughput. Finally, we include a case study where AdamA is combined with DeepSpeed ZeRO-DP to further explore the ability to train large-scale models and push the limit to reduce memory of gradients, activations and optimizer states.

### 4.1 Convergence Behavior

To verify the convergence properties of AdamA, we experiment on both NLP (transformer-based) and CV (convolution-based) models. We set the same mini-batch size when training with Adam and AdamA. We set the accumulation steps N to 2,4,8 when training with AdamA.

For NLP model, we follow RoBERTa method (Liu et al., 2019) to pre-train BERT-Large ($L = 24, H = 1024, A = 16, 340M$) on a DGX A100 with sequence length of 128 and mini-batch size of 1024. We use the implementation of BERT model from Microsoft DeepSpeed (Rasley et al., 2020).

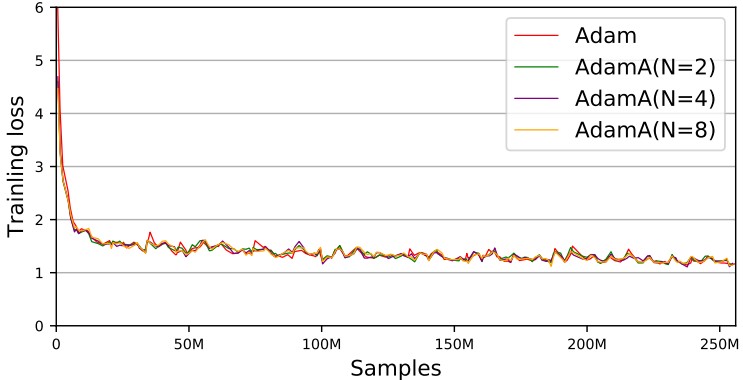

Figure 2: Sample-wise convergence properties for BERT-Large pre-training with sequence length 128 using a DGX A100. AdamA has almost the same training loss curve with Adam.

For the pre-training corpus, we use the English Wikipedia and BooksCorpus downloading from NVIDIA GitHub (NVIDIA). It should be noted that the corpus we use is different from that used in original BERT (Devlin et al., 2018) because the BERT corpus is not available to the public at this time. For the pre-training hyper-parameters, we follow the method in RoBERTa.

Table 1: Similar accuracy can be achieved by fine-tuning on the BERT-Large model pre-trained by Adam and AdamA. F1 scores are reported for MRPC, Spearman correlations are reported for STS-B, and accuracy scores are reported for other tasks.

| Setting | MNLI-M | MNLI-MM | SST-2 | MRPC | STS-B | QNLI | QQP | RTE | CoLA |
|---|---|---|---|---|---|---|---|---|---|
| Adam | **80.62** | 80.96 | 90.48 | 84.03 | **86.56** | 87.46 | **86.53** | 58.48 | 43.25 |
| AdamA (N=2) | 80.50 | 80.72 | **90.83** | **86.27** | 84.18 | **87.59** | 86.49 | 57.04 | 43.27 |
| AdamA (N=4) | 80.38 | 80.93 | 90.48 | 85.39 | 85.71 | 87.30 | 86.52 | 56.68 | 43.11 |
| AdamA (N=8) | 80.39 | **80.97** | 89.79 | 85.67 | 85.62 | 87.44 | 86.51 | **61.01** | **43.86** |

Figure 2 presents the sample-wise convergence results when training BERT-Large with Adam and AdamA. No matter how many micro-batches in one mini-batch, we find the convergence curve of AdamA coincides with that of Adam. To further evaluate the convergence of the BERT-Large model trained by Adam and AdamA, we fine-tune the models on all tasks from GLUE benchmark (Wang et al., 2018). We fine-tune for 3 epochs for all tasks and select the best fine-tuning learning rate (among 2e-5, 3e-5, 4e-5, 5e-5) on the Dev set. Table 1 shows the fine-tuning results. Obviously, the model pre-trained with AdamA provides similar accuracy with that pre-trained with Adam.

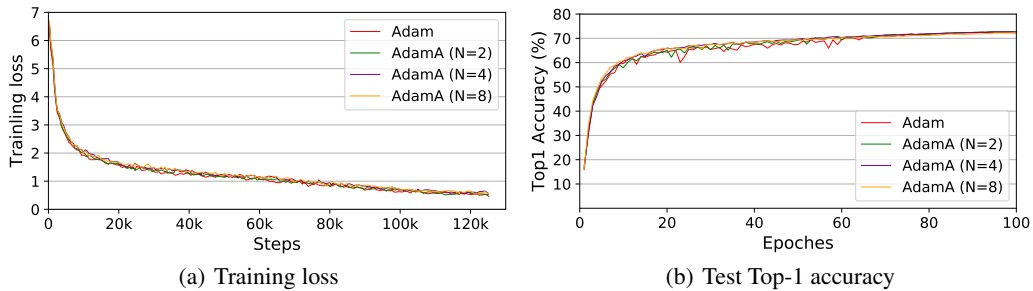

(a) Training loss          (b) Test Top-1 accuracy

Figure 3: The training loss curve and the test accuracy of ResNet-50 on ImageNet.

For CV model, we train ResNet-50 with 4 A100 GPUs on ImageNet (Deng et al., 2009) dataset to evaluate the convergence properties of AdamA. Following the training setting provided by MMClassification (mmlab), we train ResNet-50 with mini-batch size of 1024. For the learning rate, we initial

it to 1e-3 and cosine decay it to 1e-5. Figure 3 presents the training loss curve and the test accuracy of ResNet-50, from which we can jump to the conclusion that AdamA has almost the same convergence properties with that of Adam in CV tasks.

Considering that Batch Normalization (BN) (Ioffe & Szegedy, 2015) is used in ResNet, we also pay attention to the effect on model accuracy which may be brought by the difference of the micro-batch normalization statistics and that statistics of the entire mini-batch. Mentioned in (Sohoni et al., 2019), the influence of BN on model convergence tends to be constant if the micro-batch size increases above a certain extent. Therefore, we do not pay efforts to keep exactly the same BN algorithm between micro-batched training and non-micro-batched one. In the experimental results shown in Figure 3, we also find the impact on convergence can be ignored.

To help understand the difference between AdamA and Adam during training process, we make the following statistics. From the update equation $\theta_t \leftarrow \theta_{t-1} - \frac{\alpha \hat{m}_t}{\sqrt{\hat{v}_t}}$, it is clear that our adaptive scaling length differs from the standard Adam in a coefficient $\sqrt{\hat{v}_t}/\sqrt{\hat{v}_t}'$. We track the coefficient in training ResNet-50 on CIFAR-100 dataset. In Figure 4, we plot the mean value of $\sqrt{\hat{v}_t}/\sqrt{\hat{v}_t}'$ during each training step and its value range. It shows generally the coefficient keeps around 1.0 and the deviation value range is within 1%. We think the minimal deviation in each iteration might contribute to the same convergence properties of Adam and AdamA during training.

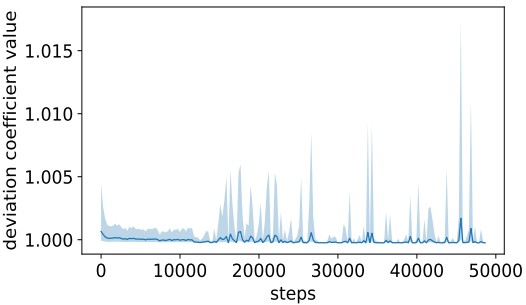

Figure 4: Statistics on the value of $\frac{\sqrt{\hat{v}_t}}{\sqrt{\hat{v}_t'}}$ generated from ResNet-50 on CIFAR-100. The real line shows the coefficient mean during iterations and the area shows its range. It shows that the value AdamA deviates from the standard Adam is within 1%.

## 4.2 MEMORY REDUCTION

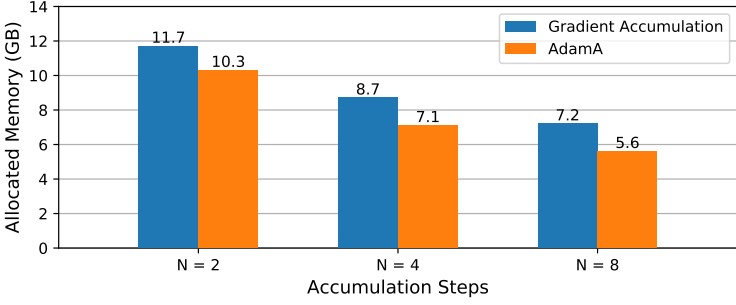

Figure 5: The memory reduction of AdamA compared with gradient accumulation when training BERT-Large.

As AdamA eliminates the contradiction when combining gradient accumulation and gradient release, we first show the improvement of AdamA compared with gradient accumulation. Compared with gradient accumulation, AdamA can save the memory footprint of both activations and gradients. We measure the memory footprint when training BERT-Large with AdamA on a DGX A100 (8 A100 GPUs) with the mini-batch size of 256 and the sequence length of 128. As shown in Figure 5, AdamA

can save 1.6GB more memory than gradient accumulation no matter how many the accumulation steps are set in a mini-batch.

Table 2: When training BERT-Large, AdamA achieves less memory usage than Adafactor (Shazeer & Stern, 2018) and SM3 (Anil et al., 2019).

| Optimizers | Reduction target | Mini-batch size per GPU | Memory usage per GPU (GB) |
|---|---|---|---|
| Adam (baseline) | N/A | 8 | 6.15 |
| Adafactor | Optimizer states | 8 | 4.83 |
| SM3 | Optimizer states | 8 | 4.90 |
| AdamA(N=8) | Activations and gradients | 8 | 4.18 |

To further show the memory saving effect of AdamA, we expand BERT model to BERT-4B with 4 billion weights using the scaling method of GPT-3 (Brown et al., 2020). We set the mini-batch size to 64 and accumulation steps to 8 in this experiment. In Figure 6(a), we train BERT-4B with gradient accumulation and AdamA using PyTorch framework. It can be found that AdamA can save 23.2% memory footprint compared with gradient accumulation when the weights number of a model get to 4 billion.

Compared with other memory-efficient optimizers, e.g. Adafactor (Shazeer & Stern, 2018) and SM (Anil et al., 2019), the memory reduction of our proposal is bigger under the same experiment setting. The comparison with Adam baseline is shown in Table 2. The reason AdamA can reach more significant memory reduction is AdamA targets at reducing the memory usage of both activations and gradients, while other works only aimed to reduce optimizer states memory. At the same time, AdamA can work well with these previous works to get further memory reduction.

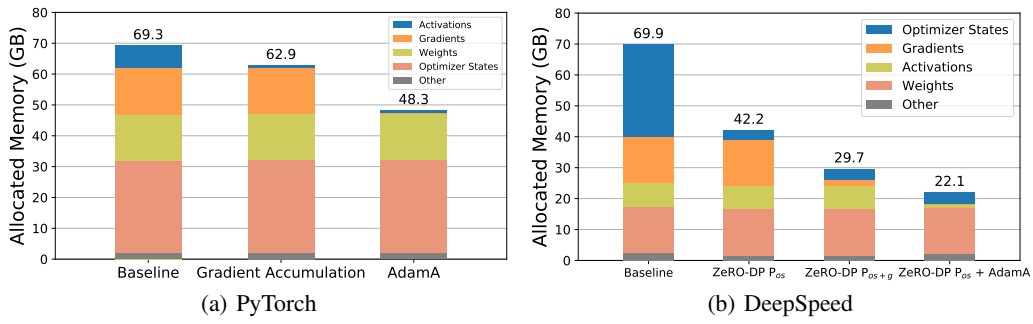

(a) PyTorch      (b) DeepSpeed

Figure 6: The memory reduction of AdamA when training BERT-4B using PyTorch and DeepSpeed.

To show the compatibility of AdamA with existing methods, we combine AdamA with ZeRO-DP (Rajbhandari et al., 2020), a popular memory reduction method for optimizer states. ZeRO-DP $P_{os}$ partitions the optimizer states to different GPUs when training with data parallelism. In Figure 6(b), we combine AdamA with ZeRO-DP $P_{os}$ to further reduce gradients and activations. It shows that AdamA with ZeRO-DP $P_{os}$ can save 20.1 GB more memory footprint than only ZeRO-DP $P_{os}$. Even compared with ZeRO-DP $P_{os+g}$, which partitions both optimizer states and gradients, our combined method can reduce 7.6 GB more memory.

Table 3: The largest model size can fit on different DGX systems with AdamA.

| | PyTorch | | DeepSpeed | |
|---|---|---|---|---|
| | Gradient accumulation | AdamA | ZeRO-DP $P_{OS}$ | ZeRO-DP $P_{OS}$ + AdamA |
| DGX-1 | 1.4B | 1.8B | 1.1B | 3.3B |
| DGX-2 | 3.0B | 4.0B | 2.5B | 6.8B |
| DGX A100 | 7.6B | 9.6B | 5.8B | 18.2B |

In Table 3, we explore the largest transformer-based model can fit on DGX systems with various memory capacity with AdamA. At present, the mainstream DGX systems on the market include DGX-1 (8 V100-16GB GPUs), DGX-2 (16 V100-32GB GPUs), and DGX A100 (8 A100-80GB GPUs). In order to keep the same experimental settings, we set the number of GPUs to 8. The mini-batch size and accumulation steps are set to 256 and 8, respectively. With PyTorch framework, the largest model AdamA can train is 1.26x to 1.33x larger than gradient accumulation can train. Combined with DeepSpeed ZeRO-DP, AdamA can train a model with 18.2 billion weights in a DGX A100, which is 3.14x larger than the model the system can train with only ZeRO-DP $P_{os}$.

## 4.3 TRAINING THROUGHPUT

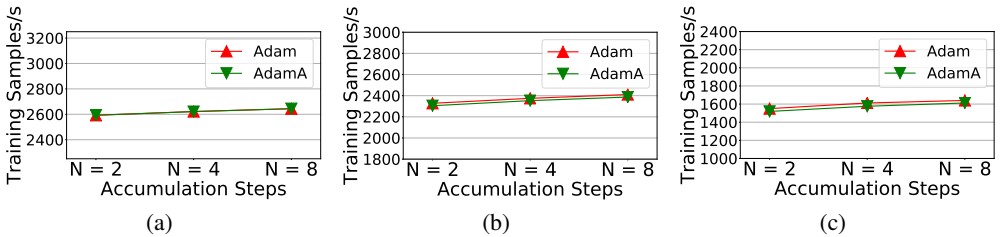

Figure 7: AdamA has less than 2% effect on the training throughput compared with gradient accumulation using Adam: (a) training ResNet-50 with single GPU; (b) training BERT-Base with 4 A100 GPUs; (c) training BERT-Large with 8 A100 GPUs.

In this section, we show AdamA has negligible impact on training throughput. During training, it is reasonable to set the micro-batch size as large as the device memory can contain, in order to saturate GPUs to achieve maximal training throughput. Therefore, the micro-batch size is fixed in this section.

**Single-GPU Scenario**    As mentioned in Section 3.3, our system design for AdamA Single-GPU implementation is intended to incur no extra throughput overhead. In Figure 7(a), we conduct a throughput comparison with standard Adam training ResNet-50 with one A100 GPU. We keep the micro-batch size to 256 and switch accumulation steps to 2, 4 and 8. We can conclude that training with AdamA has little throughput impact in single-GPU scenario.

**Distributed Data Parallel Scenario**    Explained in Section 3.3, our system design keep the communication number to be constant by synchronizing the optimizer states. Although it may incur more communication volume compared with standard Adam that synchronizes the gradient, the impact on throughput is minimal. In Figure 7(b)(c), we conduct multi-GPU experiments with two models: BERT-Base with 4 A100 GPUs and BERT-Large with 8 A100 GPUs. The micro-batch size of all the models is set to 1024. The experiments show that the training throughput difference is within 2%. The throughput gap between AdamA and Adam is gradually decreasing with the increase of gradient accumulation steps. This is because the communication volume is constant in a mini-batch, the communication overhead proportion becomes smaller in a mini-batch with larger gradient accumulation steps.

## 5 CONCLUSION

This paper presents Adam Accumulation (AdamA), a novel optimizer method for large-scale DNN training. It enables saving memory footprints of activations and gradients simultaneously. Besides, AdamA yields the same convergence properties as Adam. Compared with gradient accumulation, AdamA can reduce the memory footprint up to 23% with less than 2% degradation in training throughput. Combined with memory reduction methods for optimizer states, AdamA can fit $1.26\times$~$3.14\times$ larger models over PyTorch and DeepSpeed baseline on different DGX systems.

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

# A  CONVERGENCE PROOF

In this section, we show the convergence analysis of AdamA. Compared with proof in the original Adam paper Appendix, it's easy to see AdamA follows almost the same analysis proof with Adam. The most obvious difference is that Adam doesn't take micro-batch into consideration. Here we use symbol $N$ as the number of micro-batch in AdamA and $b$ as subscript for micro-batch index.

Here we highlight those conclusions that differs between Adam and AdamA from Adam paper Appendix.

First, we construct our proof based on the claim a convex function can be lower bounded by a hyperplane, which is Mathematically expressed in Lemma 2.

**Definition 1.** A function $f : R_d \to R$ is convex if for all $x, y \in R^d$, for all $\lambda \in [0, 1]$,

$$\lambda f(x) + (1 - \lambda)f(y) \geq f(\lambda x + (1 - \lambda)y)$$

**Lemma 2.** If a function $f : R^d \to R$ is convex, then for all $x, y \in R^d$,

$$f(y) \geq f(x) + \nabla f(x)^T (y - x)$$

Lemma 3 and Lemma 4 are proved to support the proof of Theorem 5.

**Lemma 3.** Let $g_t = \nabla f_t(\theta_t)$ and $g_{1:t}$ be defined as above and bounded,$\|g_t\|_2 \leq G$, $\|g_t\|_\infty \leq G_\infty$. Then.

$$\sum_{t=1}^{T} \sqrt{\frac{g_{t,i}^2}{t}} \leq 2G_\infty \|g_{1:T,i}\|_2$$

*Proof.* The proof is the same as Lemma 10.3 in Adam Appendix(Kingma & Ba, 2014) and hence is ommitted here.

**Lemma 4.** Let $\gamma \triangleq \frac{\beta_1^2}{\sqrt{\beta_2}}$. For $\beta_1, \beta_2 \in [0, 1)$ that satisfy $\frac{\beta_1^2}{\sqrt{\beta_2}} < 1$ and bounded $g_t$, $\|g_t\|^2 \leq G$, $\|g_t\|_\infty \leq G_\infty$, and the micro-batch number equals to $N$, the following inequality holds

$$\sum_{t=1}^{T} \frac{\hat{m}_{t,i}^2}{\sqrt{t\hat{v}_{t,i}}} \leq \frac{2}{1 - \gamma} \frac{1}{\sqrt{1 - \beta_2}} \sum_{b=1}^{N} \|g_{1:T,i,\,b}\|_2$$

*Proof.*

$$\sum_{t=1}^{T} \frac{\hat{m}_{t,i}^2}{\sqrt{t\hat{v}_{t,i}}} = \sum_{t=1}^{T-1} \frac{\hat{m}_{t,i}^2}{\sqrt{t\hat{v}_{t,i}}} + \frac{\sqrt{1-\beta_2^T}}{(1-\beta_1^T)^2} \frac{(\sum_{k=1}^{T}(1-\beta_1)\beta_1^{T-k}g_{k,i})^2}{\sqrt{T\sum_{j=1}^{T}(1-\beta_2)\beta_2^{T-j}g_{j,i}^2}}$$

$$\leq \sum_{t=1}^{T-1} \frac{\hat{m}_{t,i}^2}{\sqrt{t\hat{v}_{t,i}}} + \frac{\sqrt{1-\beta_2^T}}{(1-\beta_1^T)^2} \sum_{k=1}^{T} \frac{((1-\beta_1)\beta_1^{T-k}g_{k,i})^2}{\sqrt{T(1-\beta_2)\beta_2^{T-k}g_{k,i}^2}}$$

$$\leq \sum_{t=1}^{T-1} \frac{\hat{m}_{t,i}^2}{\sqrt{t\hat{v}_{t,i}}} + \frac{\sqrt{1-\beta_2^T}}{(1-\beta_1^T)^2} \frac{(1-\beta_1)^2}{\sqrt{T(1-\beta_2)}} \sum_{k=1}^{T} T\left(\frac{\beta_1^2}{\sqrt{\beta_2}}\right)^{T-k} \|g_{k,i}\|_2$$

$$= \sum_{t=1}^{T-1} \frac{\hat{m}_{t,i}^2}{\sqrt{t\hat{v}_{t,i}}} + \frac{\sqrt{1-\beta_2^T}}{(1-\beta_1^T)^2} \frac{(1-\beta_1)^2}{\sqrt{T(1-\beta_2)}} \sum_{k=1}^{T} T\left(\frac{\beta_1^2}{\sqrt{\beta_2}}\right)^{T-k} \|\sum_{b=1}^{N} g_{k,i,\,b}\|_2$$

$$\leq \sum_{t=1}^{T-1} \frac{\hat{m}_{t,i}^2}{\sqrt{t\hat{v}_{t,i}}} + \frac{\sqrt{1-\beta_2^T}}{(1-\beta_1^T)^2} \frac{(1-\beta_1)^2}{\sqrt{T(1-\beta_2)}} \sum_{k=1}^{T} T\left(\frac{\beta_1^2}{\sqrt{\beta_2}}\right)^{T-k} \sum_{b=1}^{N} \|g_{k,i,\,b}\|_2$$

$$\leq \sum_{t=1}^{T-1} \frac{\hat{m}_{t,i}^2}{\sqrt{t\hat{v}_{t,i}}} + \frac{T}{\sqrt{T(1-\beta_2)}} \sum_{k=1}^{T} \gamma^{T-k} \sum_{b=1}^{N} \|g_{1:T,i,\,b}\|_2$$

$$\leq \sum_{t=1}^{T} \frac{\sum_{b=1}^{N} \|g_{1:T,i,\,b}\|_2}{\sqrt{t(1-\beta_2)}} \sum_{j=0}^{T-t} t\gamma^j$$

$$\leq \sum_{t=1}^{T} \frac{\sum_{b=1}^{N} \|g_{1:T,i,\,b}\|_2}{\sqrt{t(1-\beta_2)}} \sum_{j=0}^{T} t\gamma^j$$

Applying Lemma 3,

$$\sum_{t=1}^{T} \frac{\hat{m}_{t,i}^2}{\sqrt{t\hat{v}_{t,i}}} \leq \frac{2G_\infty}{(1-\gamma)^2\sqrt{(1-\beta_2)}} \sum_{b=1}^{N} \|g_{1:T,i,\,b}\|_2$$

$\square$

**Theorem 5.** Assume that the function $f_t$ has bounded gradients, $\|\nabla f_t(\theta)\| \leq G$, $\|\nabla f_t(\theta)\|_\infty \leq G_\infty$ for all $\theta \in R^d$ and distance between any $\theta_t$ generated by AdamA is bounded, $\|\theta_n - \theta_m\|_2 \leq D$, $\|\theta_n - \theta_m\|_\infty \leq D_\infty$ for any $m, n \in \{1, ..., T\}$, and $\beta_1, \beta_2 \in [0, 1)$ satisfy $\frac{\beta_1^2}{\sqrt{\beta_2}} < 1$, and the micro-batch number equals to $N$. Let $\alpha_t = \frac{\alpha}{\sqrt{t}}$ and $\beta_{1,t} = \beta_1 \lambda^{t-1}, \lambda \in (0, 1)$.

AdamA achieves the following guarantee, for all $T \geq 1$.

$$R(T) \leq \frac{D^2}{2\alpha(1-\beta_1)} \sum_{i=1}^{d} \sqrt{T\hat{v}_{T,i}} + \frac{\alpha(\beta_1+1)G_\infty}{(1-\beta_1)\sqrt{1-\beta_2}} \sum_{i=1}^{d} \sum_{b=1}^{N} \|g_{1:T,i,\,b}\|_2 + \sum_{i=1}^{d} \frac{D_\infty^2 G_\infty \sqrt{1-\beta_2}}{2\alpha(1-\beta_1)(1-\lambda)^2}$$

*Proof.*

Following the proof in Adam Appendix Theorem 10.5, by applying Lemma 4, we can get the final convergence bound by substituting $\sum_{i=1}^{d} \sum_{b=1}^{N} \|g_{1:T,i,\,b}\|_2$ for $\sum_{i=1}^{d} \|g_{1:T,i}\|_2$. $\square$

Thus, it can be claimed that the convergence rate $\frac{R(T)}{T} = O(\frac{1}{\sqrt{T}})$, the same as Adam does.

## B   MORE DETAILS ABOUT ADAMA IN THE DISTRIBUTED DATA PARALLEL SCENARIO

In the distributed data parallel scenario, we pay efforts to make the update effect of $m_t$ and $v_t$ of AdamA (number of GPUs = M, number of microbatches per minibatch = N) consistent with the

update effect of AdamA (number of microbatches per minibatch = NM) in single device scenarios. To achieve the effect, we propose a new update method for m and v among different GPUs.

As mentioned in Section 3.3, we choose to all-reduce optimizer states instead of gradients at the end of each mini-batch. In this way, the value of $m_t$ and $v_t$ are shown below before optimizer states are all-reduced. M equals to the number of GPUs, and N equals to the number of microbatches per minibatch. Other symbols follow our definition in Algorithm 1, and $g_{t,i} \leftarrow \frac{1}{N}\nabla_\theta f_{t,i}(\theta_{t-1})$. Notice that we will multiply $v_{t-1}$ by $M\beta_2$ instead of $\beta_2$ before the start of each minibatch.

$$m_t = \beta_1 m_{t-1} + (1-\beta_1)\sum_{i=0}^{N-1} g_{t,i} = \beta_1 m_{t-1} + (1-\beta_1)\sum_{i=0}^{N-1}\frac{\nabla_\theta f_{t,i}(\theta_{t-1})}{N}$$

$$v_t = M\beta_2 v_{t-1} + (1-\beta_2)\sum_{i=0}^{N-1} g_{t,i}^2 = M\beta_2 v_{t-1} + (1-\beta_2)\sum_{i=0}^{N-1}(\frac{\nabla_\theta f_{t,i}(\theta_{t-1})}{N})^2$$

During all-reduce operations for $m_t$, we take the average of $m_t$ from each GPU (add them together and divide by M). For $v_t$, we divide by $M^2$ instead of M after summing $v_t$ from each GPU. After that, the value of $m_t$ and $v_t$ in each GPU are:

$$m_t = \beta_1 m_{t-1} + (1-\beta_1)\sum_{i=0}^{NM-1}\frac{\nabla_\theta f_{t,i}(\theta_{t-1})}{NM}$$

$$v_t = \beta_2 v_{t-1} + (1-\beta_2)\sum_{i=0}^{NM-1}(\frac{\nabla_\theta f_{t,i}(\theta_{t-1})}{NM})^2$$

It is easy to find the $m_t$ and $v_t$ keep consistent with the $m_t$ and $v_t$ from Algorithm 1, as long as we replace N with NM in line 5 "$g_{t,i} \leftarrow \frac{1}{N}\nabla_\theta f_{t,i}(\theta_{t-1})$". In this way, we make the update effect of $m_t$ and $v_t$ in distributed scenarios (number of GPUs = M, number of microbatches per minibatch = N) consistent with the update effect of AdamA in single device scenario (number of microbatches per minibatch = NM). As the convergence properties of AdamA has been proven the same with Adam in single device scenarios, its convergence properties can also be guaranteed in distributed scenarios.

