# OpenReview forum: "Adam Accumulation to Reduce Memory Footprints of both Activations and Gradients for Large-scale DNN Training"
_ICLR.cc/2023/Conference — Submitted to ICLR 2023_

### Official Review · Reviewer_a8MG · 2022-10-23

**Confidence:** 3
**Correctness:** 3
**Technical Novelty And Significance:** 4
**Empirical Novelty And Significance:** 3
**Recommendation:** 6

**Clarity, Quality, Novelty And Reproducibility:**

Clarity and reproducibility of the paper is good. The paper gives details and sources of the experiments and also provide source code to reproduce. Overall writing quality is good as well: easy to follow the main claims and their supports.
Novelty of the paper: to the best of my knowledge (however I am not expert in memory reduction in DNN), the method of intergrating the gradient accumulation into the optimier states is novel. The claimed contributions around this method are mostly well-supported (as above mentioned). So the novelty of the paper is good in my opinion.

**Details Of Ethics Concerns:**

Not applicable.

**Strength And Weaknesses:**

Strength:
1. novelty of the proposed method: to the best of my knowledge, this method is novel and adds quite significant value on top of the baseline gradient accumulation method, in terms of memory reduction. It enables combining benefits of different memory reduction methods.

2. the contributions are mostly well-supported.

a) [well-supported] the claimed contribution of the proposed AdamA can enable reducing memory footprints of activations and gradients simultaneously, leading to good amount of memory reduction compared to previous work. This claim is well-supported by multiple empirical evidences, including directly comparing memory allocation and the max model size on both common vision and language datasets/models. Examples include ResNet-50 and Bert-large.

b) [well-supported] the proposed method in parallel training setting could have suffered from heavy communication overhead due to the micro-batch all-reduce operation. The authors proposed to address this by updating local optimizer states without micro-batch all-reduce and update once at the end of mini-batch. This method made the throughput of the propoesd method to be similar to original Adam method. The empirical results supports the claim.

c) [still have concerns] the authors demonstrate some evidence that the proposed method has similar convergence as Adam: mathmatically analysis about the convergence and empirically showing similar performance on ImageNet and GLUE. However these evidence may not be sufficient, plesae see the weakness part.

3. clarity, reproducibility, quality are good (as mentioned in the next question).


Weakness (or unaddressed concerns):

The proposed method made two major changes compared to Adam: one is the 2nd momentum accumulation in the optimizer states by replacing the square of the accumulated gradients to accumulating the square of gradients. the other one is in parallel setting, replacing the all-reduce per micro-batch with local update per micro-batch. These two intuitively may have impact on model performance/convergence, and thus need evidence to show their impact is negligible for the proposed method to be useful.

While the authors have shown some evidences, I still have some concerns unaddressed:

1. the convergence analysis is done for the first change, but not the 2nd change (even though it only happens with parallel setting, but I think large-scale DNN basically means we use parallel setting all the time). Also the analysis is based on non-pratical assumptions, such as convex cost function. I understand non-convex scenarios are usually not provided in such analysis. But I do want to call out this weakness to emphasize that we would need strong empirical evidence to support the "similar convergence" claim.

2. The empirical evidence of convergency versus memory reduction misses some important results. The authors separately shows the convergence results in Sec 4.1 and memory reduction in Sec 4.2. However for some experiments settings, we can only see one but not the other (e.g. we have memory reduction results of Adafactor etc but not its convergence/performance). Could the authors provide the full comparison (note this should not require new experiments, just need all the metrics that should have been obtained during authors' original experiments)? Specifically, examples could be a table of <method, memory reduction, accuracy metrics> and a table of <method, max model size, accuracy metrics>, on top of table 2 and table 3.

3. The evidence shown on the vision domain is not very strong. The authors used ResNet-50 with Adam as baseline. However we know that Adam is much weaker that SGD on ResNet-50 and can only achieve sub-optimal performance on ImageNet (fig. 3 shows <75% but SGD gives more than 75% I remember). Thus ResNet-50 is not a good model to compare AdamA and Adam. If the author would like to strengthen the claim on vision domain, I would recommend to use some models that originally uses Adam as its strongest optimizer, like transformer based models.

**Summary Of The Paper:**

This paper focus on the problem of reducing memory usage in parallel large-scale DNN training. The problem itself is indeed an important research problem that can benefits both the research community and industry. The authors proposed a method (AdamA) to improve upon previous memory reduction methods. The method is that: during gradient accumulation over multiple micro-batch, we can directly integrates gradients into optimizer states and release the gradients.

From emipirical study, the authors demonstrated 1) AdamA can further reduce memory compared to previous work, on various settings. 2) AdamA shows similar convergency as Adam baseline. Authors also provide a mathmatical analysis for the convergence. In addition, the authors provide a easy to apply training pipeline that can be used by other researchers.

**Summary Of The Review:**

As mentioned above, I think the quality of the paper, the novelty, and most of the evidence for supporting the claimed contributions are quite good. My major concern lies on the convergence part. For the proposed method to be pratically helpful to the research community and industry, this convergence part needs to strongly supported. If the author can provide explanantions and evidence to address my concerns, I would be happy to accept the paper.

---

> ### Author Response · Authors · 2022-11-09
> **Response to Reviwer a8MG**
>
> **Q**
>
> “The convergence analysis is done for the first change, but not the 2nd change. The convergence analysis is based on non-convex scenarios, which is non-pratical assumptions.”
>
>
>
> **A**
>
> We didn’t show the convergence analysis for the 2nd change because we have made the update effect of $m_t$ and $v_t$ of distributed AdamA (number of GPUs = M, number of microbatches per minibatch = N) consistent with the update effect of AdamA (number of microbatches per minibatch = NM) in single device scenarios. As the convergence properties of AdamA have been proven similar with Adam in single device scenarios, its convergence properties can also be guaranteed in distributed scenarios. Thank you for your suggestions, we revised our paper and the detailed explanation has been attached at the end of the appendix.
>
>
>
> As you can see, non-convex scenarios are usually not provided in such analysis. However, we believe non-convex scenarios should be token into consideration if the proof needs to be rigorous. We look forward to the entire community working with us to improve this type of work in the future.
>
> $\newline$
>
> **Q**
>
> "...misses accuracy comparison in table2 and table3"
>
>
>
> **A**
>
> Our idea is to reduce memory footprint as well as to align accuracy between AdamA and Adam. We list Adafactor and SM3 in table 2 to compare with similar work in terms of memory-saving capabilities, and to lead to the discussion that AdamA can be combined with other methods.
>
>
>
> Following your suggestions, we append the Masked LM test accuracy of each optimizer after training 500k steps in the table below. The influence of SM3 and AdamA on the accuracy can be consistent with Adam, but Adafactor has bad impacts on the accuracy. These conclusions are consistent with those in [1].
>
>
>
> | Optmizers       | Reduction target          | Mini-batch size per GPU | Memory per GPU (GB) | Masked LM test accuracy |
> | --------------- | ------------------------- | ----------------------- | ------------------- | ----------------------- |
> | Adam (baseline) | N/A                       | 8                       | 6.15                | 69.7%                   |
> | Adafactor       | Optimizer states          | 8                       | 4.83                | 66.2%                   |
> | SM3             | Optimizer states          | 8                       | 4.90                | 69.6%                   |
> | AdamA (N=8)     | Activations and gradients | 8                       | 4.18                | 69.7%                   |
>
>
>
> Regarding Table3, since we only want to show our memory-saving ability, we only trained a few epochs to give the memory cost, and there is no complete accuracy data. We also consider training these models according to your suggestions, but the total training time is intolerably long. For example, we estimate that it will take more than 500 days to complete training the model of 18.2B. We cannot afford it at this moment, please understand.
>
> $\newline$
>
> **Q**
>
> "For vision tasks, I recommend using some models that originally uses Adam as its strongest optimizer, like transformer based models."
>
>
>
> **A**
>
> The original intention of our experiment setup is to prove that AdamA is effective in both the transformer-based and the convolution-based networks. In this paper, we choose the two most representative networks: BERT-Large (transformer-based) and ResNet-50 (convolution-based).
>
>
>
> We also listen to your suggestions. We are training ViT-B-16 (the optimizer used in the original paper is Adam, 86M parameter). Due to the long training time, only the first 50 of the 300 training epochs have been completed. At the same time, we also trained more large networks of convolution, ResNet-101 (original optimizer is SGD, 43M), EfficientNet-B7 (original optimizer is RMSProp, 66M).
>
>
>
> The models are all trained on ImageNet dataset with Adam and AdamA (N=8, N is the number of micro-batches in one mini-batch). Other hyperparameters (e.g. mini-batch size and learning rate) remain the same for each model.
>
>
>
> For ViT-B-16, we train the model with Adam and AdamA for only 50 epochs (300 epochs are needed in ViT paper). Adam baseline reaches 55.20% top-1 accuracy, and AdamA (N=8) reaches 56.78% top-1 accuracy. For ResNet-101, Adam baseline reaches 75.43% top-1 accuracy, and AdamA (N=8) reaches 75.39% top-1 accuracy. For EfficientNet-B7, Adam baseline reaches 81.32% top-1 accuracy, and AdamA (N=8) reaches 81.43% top-1 accuracy.
>
> $\newline$
>
> [1] Rohan Anil, Vineet Gupta, Tomer Koren, and Yoram Singer. Memory efficient adaptive optimization. Advances in Neural Information Processing Systems, 32, 2019.

---

> ### Author Response · Authors · 2022-12-07
> **A gentle reminder**
>
> Dear Reviewer a8MG,
>
> As the end of the discussion period is coming (Dec. 12), we are wondering if your concerns have been addressed by our responses. If you have additional concerns or questions please let us know. We are happy to answer them.
>
> Thank you,
>
> Authors

---

### Official Review · Reviewer_3RgH · 2022-10-23

**Confidence:** 5
**Correctness:** 3
**Technical Novelty And Significance:** 2
**Empirical Novelty And Significance:** 2
**Recommendation:** 3

**Clarity, Quality, Novelty And Reproducibility:**

The presentation is clear because of the simplicity of the methodology.

This is a mismatch of AdamA's goal and the empirical study, especially for the distributed scenario. Why is the benchmark based on the combination with ZeRO-DP instead of more memory-efficient stages in ZeRO to support larger-scale models (where the memory reduction technique does matter)? For example, ZeRO-S3 (which is also known fully sharded data parallelism)? Can AdamA be combined with ZeRO-S3?

The source code is attached to provide good Reproducibility.

**Strength And Weaknesses:**

Strength:

- The idea is simple and direct, it is clear that the proposed method can reduce the memory footprint.

- The presentation of the paper is clear because of the simplicity of the methodology.

Weakness:

- There is a lack of more direct alternatives, e.g.,  why not simply use a smaller batch? Would AdamA be better than small batch in any case?

- AdamA introduces 2X communication overhead compared with the standard data parallelism.

- The experiments are inappropriately designed in the distributed scenario.



**Summary Of The Paper:**

This paper proposes AdamA, a memory-efficient version of Adam when combined with gradient accumulation. The idea is simple: instead of accumulating the gradients after each forward and backward step for the micro-batch, AdamA directly updates Adam optimizer states so that the memory footprint for the gradients will be reduced.

**Summary Of The Review:**

The idea is simple, straightforward but with limited novelty; some discussion about more direct alternatives is missing (e.g., why not directly use a smaller batch?). Additionally, AdamA actually introduces more communication overhead (running allreduce over m and v introduces 2X communication compared to allreduce over g).

---

> ### Author Response · Authors · 2022-11-10
> **Response to Reviewer 3RgH**
>
> **Q**
>
> “why not simply use a smaller batch? Would AdamA be better than a small batch in any case?”
>
> **A**
>
> We think comparing with simply using a smaller batch size is unnecessary because using a smaller batch has its own drawbacks and actually our study has already compared AdamA with gradient accumulation, which is an optimized alternative for simply reducing batch size. Specifically,
>
> 1) As we all know, too small batch size may cause negative impact on model convergence. Gradient accumulation can address this issue by splitting a big mini-batch into a sequence of micro-batches and accumulates their gradients. Gradient accumulation has the same memory reduction effect as simply using a smaller batch, while maintaining the same convergence as using a big mini-batch. Apperently, gradient accumulation is a stronger and more meaningful baseline than simply using a smaller batch size.
>
> 2) Directly using a smaller batch size or gradient accumulation can only reduce the memory footprint of activations, while other parts (e.g., weights, gradients, optimizer states) could dominate the memory occupation in different models or training settings. Our study enables saving both activation and gradient memory. Thus, it improves the gradient accumulation method and is better than simply using a smaller batch size.
>
> $\newline$
>
> **Q**
>
> "AdamA introduces 2X communication overhead"
>
> **A**
>
> As the throughput is affected by both computation time and communication time, we should analyze their respective proportion of total training time and tell the effect on throughput. Our experiments prove that the throughput overhead is only 2% in actual training. Please read Section 4.3.
>
> Our goal is to train larger models with minimum trade-off overhead and we achieved it. In data parallel training, we strive to optimize the original N times overhead to a fixed 2 times. Please read Section 3.3.
>
> $\newline$
>
>
> **Q**
>
> "The experiments are inappropriately designed in the distributed scenario. Why not combine with ZeRO-S3 to support large-scale models?"
>
> **A**
>
> We can combine AdamA with ZeRO-S3 because our method to reduce activation and gradient is orthogonal to ZeRO-S3’s weight sharding. In figure 6(b) settings, the memory can be reduced to 9.1 GB if we further combine AdamA with ZeRO-S3.
> In practice, the most common stage researchers use is ZeRO-S1 because it has limited impact on training throughput. Therefore, we chose to combine AdamA with ZeRO-S1.
>
> $\newline$
>
>
> **Q**
>
> Concerns about the simplicity and novelty
>
> **A**
>
> The gradient release and gradient accumulation methods save gradients and activations memory respectively, but these two are inherently mutually exclusive. There are many challenges to combine the two including: 1) the convergence behavior of the new optimizer needs to be analyzed; 2) a new training process is required; 3) the big communication overhead in multi-GPU training needs to be reduced. Due to these challenges, no one has ever tried to combine the two in the past but chose to use one or the other.  AdamA overcomes these obstacles with a simple but effective method, and for the first time enables saving both activation and gradient memory.
>
>
> More importantly, AdamA can benefit the entire community and point out a memory-efficient method for other optimizers: our optimizer accumulation method is likely to be effective for memory reduction while maintaining convergence on other optimizers. We think our work is a bold and successful attempt in this field.

---

> ### Author Response · Authors · 2022-12-07
> **A gentle reminder**
>
> Dear Reviewer 3RgH,
>
> As the end of the discussion period is coming (Dec. 12), we are wondering if your concerns have been addressed by our responses. If you have additional concerns or questions please let us know. We are happy to answer them.
>
> Thank you,
>
> Authors

---

### Official Review · Reviewer_6Jc2 · 2022-10-30

**Confidence:** 3
**Correctness:** 2
**Technical Novelty And Significance:** 3
**Empirical Novelty And Significance:** 4
**Recommendation:** 6

**Clarity, Quality, Novelty And Reproducibility:**

Clarity: The paper is quite well-written. It explains the previous approaches to reduce memory usage quite well, and places the proposed work properly in that context. Several figures, diagrams and detailed algorithm descriptions make the approach quite easy to understand.

Originality: The paper borrows quite a lot from previous approaches, but as the experimental results demonstrate the proposed modifications lead to significant improvements in practice.

Minor question:

Can the authors comment on the running time of AdamA, as compared to Adam, in some of the considered settings?

**Strength And Weaknesses:**

Strengths:

1. The paper provides a simple and clean approach to reduce memory consumption for neural network training. The idea is compatible with several other approaches for reducing memory usage, and could be composed with other methods to yield even more savings.
2. The experimental evidence appears to be quite thorough. The experiments are systematic, and performed over a variety of vision and NLP tasks. They demonstrate that AdamA has very similar convergence to Adam, while providing substantial memory savings.

Weaknesses:

1. The paper proves convergence of AdamA by following the proof of convergence for Adam. Unfortunately, the proof of convergence for Adam has fatal flaws as pointed out by "On the Convergence of Adam and Beyond". I tried to go through the proof, and I think the proof in the paper inherits the same flaws. In particular, "On the Convergence of Adam and Beyond" shows that Adam in fact does not always converge, and the same counterexample should apply to AdamA as well. It is possible that some of proposed variants to rectify this convergence issue (such as AMSGrad) can be adapted to this setup as well, but it would have to be analyzed.

To be clear, I do not believe that the theoretical properties are the selling point of the approach. Instead, the selling point is the strong empirical results, which appear to be sound as far as I can see.

**Summary Of The Paper:**

The paper considers the problem of reducing memory usage for training neural networks. It proposes a memory-efficient version (AdamA) of the popular Adam optimizer. The motivation behind AdamA is to enable gradients to be released after computation over each micro-batch (micro-batching computes the gradient sequentially over a mini-batch to reduce the memory footprint of storing the activations over the mini-batch, while still maintaining the convergence properties of the original mini-batch). To enable the gradients to be released, AdamA incorporates the gradients into the optimization state (m_t and v_t for Adam). Allowing for gradient release enables AdamA to use prior work which computes the backward pass in a layer by layer manner, releasing the gradients for previous layers. By using both micro-batching and gradient release, AdamA can lead to considerable memory reductions.

The paper empirically validates that AdamA has similar convergence properties as Adam on several benchmarks, while being able to reduce the memory usage by order of a few GBs.

The paper also provides a proof that AdamA has similar convergence behavior as Adam, though I have some issues with that part.



**Summary Of The Review:**

In summary, though the theory in the paper appears to be flawed and this should be addressed, I don't see that as being a fatal flaw. I think the paper could lead to improvements in neural network training and I tend towards acceptance.

---

> ### Author Response · Authors · 2022-11-09
> **Response to Reviewer 6Jc2**
>
> **Q**
>
>
>
> "Adam in fact does not always converge, and the same counterexample should apply to AdamA as well"
>
>
>
> **A**
>
>
>
> Thank you for your remind. In the paper "on the convergence of adam and beyond", the authors suggests that Adam could lead to undesirable behavior because of the violation of positive definiteness for the adaptive scaling term. Hence AMSGrad optimizer is proposed.
>
>
>
>
>
> We understand that AdamA may face similar convergence issue because AdamA adapts from Adam. **The core of our method AdamA originates from the dilemma between the second order term and microbatch procedure.** In this paper, we target at Adam optimizer because it is one of the most popular optimizers in practice. Besides, our idea of optimizer accumulation is likely to be effective on other optimizers, e.g. AMSGrad. Likewise, we can also prove the convergence of "AMSGrad Accumulation":
>
>
>
> $$R_T \leq \frac{D^2_\infty \sqrt{T}}{\alpha(1-\beta_1)}\sum_{i=1}^d \hat{v}_{T,i}^{1/2} + \frac{\beta_1 D^2_\infty G_\infty}{(1-\beta_1)^2(1-\lambda)^2}$$
>
>
> $$+ \frac{\alpha\sqrt{1+logT}}{(1-\beta_1)^2(1-\gamma)\sqrt{(1-\beta_2)}} \sum_{i=1}^d\sum_{b=1}^N 2\|g_{1:T,i,b}\|_2$$
>
>
>
>
>
> To summarize, the theory flaw comes from original Adam. And our intention is not to propose a new optimizer but rather to enable a more effective memory reduction method. Besides, our method is likely to be applicable to other momentum based adaptive scaling optimizers.
>
>
> $\newline$
>
>
> **Q**
>
>
>
> "Can the authors comment on the running time of AdamA, as compared to Adam, in some of the considered settings?"
>
>
>
> **A**
>
>
>
> Sure. Here is some data from our experiments:
>
>
>
> When we trained ResNet-50 with one A100 GPU, the total running time for Adam and AdamA (N=8) are both 13.4h.
>
>
>
> When we trained BERT-Base with 4 A100 GPUs, the total  running time for Adam and AdamA (N=8) are 29.5h and 29.8h, respectively.
>
>
>
> When we trained BERT-Large with 8 A100 GPUs, the total  running time for Adam and AdamA (N=8) are 43.4h and 44.1h, respectively.

---

> ### Author Response · Authors · 2022-12-07
> **A gentle reminder**
>
> Dear Reviewer 6Jc2,
>
> As the end of the discussion period is coming (Dec. 12), we are wondering if your concerns have been addressed by our responses. If you have additional concerns or questions please let us know. We are happy to answer them.
>
> Thank you,
>
> Authors

---

### Official Review · Reviewer_u5im · 2022-11-07

**Confidence:** 4
**Clarity, Quality, Novelty And Reproducibility:** See above.
**Correctness:** 2
**Technical Novelty And Significance:** 2
**Empirical Novelty And Significance:** 2
**Recommendation:** 3

**Strength And Weaknesses:**

Overall, the optimization makes sense, and why the optimization delivers improved performance in the author's benchmarks is understandable. It's also useful to know that this modification to Adam *can* be done without impairing its convergence qualities.

Unfortunately, from my understanding of this paper, it seems to be of limited applicability. In particular, AdamA *cannot* be combined with Zero-DP P_{os} (i.e. optimizer state sharding) and gradient accumulation without significantly increasing communication costs.

To summarize my understanding of AdamA, the primary idea is that it modifies the optimizer state update to allow for online updates of the optimizer state. Thus, instead of needing to compute all of your gradient minibatches before updating the optimizer state, you can update the optimizer state (and thus release the gradient minibatches) before all of your gradient minibatches are completed. *Note* that this does require a change to how data-parallel training is typically done. Typically, data-parallel training accumulates your gradients and then all-reduces your gradients across all devices. However, as AdamA would like to release your gradients as soon as possible, AdamA must all-reduce the *optimizer state* across all devices after finishing microbatches. This is a typically an increase in communication volume, as gradients are only one value while optimizer states (i.e. of Adam) are two floating point values.

To review Optimizer State Sharding, the idea is that instead of replicating *both* your parameters and your optimizer state across all devices (like typical data parallelism), you only replicate your parameters, and shard your optimizer state across your devices. Then, after you compute your gradients, you can perform a reduce-scatter communicate your gradients to the devices that need them (as only the device with the optimizer state of layer N needs the gradients for layer N), perform your parameter update, and then perform an all-gather to replicate your parameters across all devices again. This, notably, is the *same* communication volume as regular data-parallelism (which only performs an all-reduce), as both reduce-scatters and all-gathers are half the communication volume of an all-reduce. Also, optimizer state sharding does not interfere with gradient accumulation.

Now, the issue here is that, from my understanding, one cannot combine all 3 of optimizer state sharding, AdamA, *and* gradient accumulation without a significant increase in communication volume.

The way that AdamA avoids communication after each minibatch in gradient accumulation is by doing an online update of the optimizer state. *However*, this can only be done as long as your optimizer state is present on every device, which is not true in Optimizer State Sharding.

Thus, although the paper presents memory reduction in Figure 6b through combining AdamA with optimizer state sharding, I suspect that the training throughput is substantially decreased.

OTOH, from Figure 6a, we can see that although it reduces the gradient memory, the bulk of the memory is *optimizer states*. So in that setting, I suspect that using optimizer state sharding would reduce the memory more compared to AdamA.

The rest of the paper seems clear enough to me, but this is my primary issue with the paper. **As my review heavily relies upon my understanding of this point, if I've misunderstood the paper (or am missing other considerations), please let me know and I would be happy to adjust my rating.**

### Conclusion

The authors claim that "AdamA can reduce the memory footprint up to 23% with less than 2% degradation in training throughput [compared to Adam baseline without other memory optimization techniques]" and that "AdamA can fit 1.26x larger models over ... Deepspeed baselines". However, from my understanding of the paper, it seems that AdamA cannot *also* fit 1.26x larger models over Deepspeed baselines without a substantial degradation in training throughput. Thus, it seems to me that I don't see a situation where one would use AdamA over optimizer state sharding.




**Summary Of The Paper:**

This paper provides a modification to Adam that allows for gradient updates to be applied "online" during gradient accumulation instead of after gradient accumulation has finished. The main selling point of this approach is that it allows for the memory savings from what the authors call "gradient release" (also equivalent to the optimization in Zero-2 or Zero-DP P_{g}) to be combined with the memory savings from gradient accumulation.

The authors demonstrate that this can lead to increased memory savings compared to Zero-DP P_{os+g}, and negligible throughput drop compared to gradient accumulation.

**Summary Of The Review:**

See above.

---

> ### Author Response · Authors · 2022-11-08
> **Response to Reviewer u5im (1)**
>
> **Q**:
>
> Limited applicability because AdamA cannot be combined with Zero-DP P_{os} (i.e. optimizer state sharding) and gradient accumulation without significantly increasing communication costs
>
> **A**:
>
> There are many scenarios where memory reduction is priority, in which ZeRO-DP alone is not enough. Here AdamA comes to the rescue. The throughput is endurable after we trade communication cost for more memory reduction. Please let us further explain it.
>
> The memory footprint during model training can be categorized into four parts: weights, gradients, optimizer states and activations. As different models, optimizers, or batch sizes lead to different ratios of the four parts, no method is optimal in all cases. For example, activations account for more than 50% in many cases. In these scenarios, activations are the bottleneck of the overall memory cost, which cannot be solved by ZeRO-DP. In addition, ZeRO-DP is not applicable in single-GPU scenarios.
>
> When training large models in DP scenarios, the communication and memory cost need to be traded off against each other. We show the total communication volumes and memory footprint for each GPU below. We follow the ZeRO-DP paper to set $\psi$  elements as the model size. For simplicity, we call ZeRO-DP P_os, P_os+g, P_os+g+w as ZeRO-s1, ZeRO-s2, and ZeRO-s3 respectively. In the tables, N is the number of microbatches in each minibatch, M is the number of GPUs, and L is the number of layers of the model.
>
> Table1: Total communication volumes for each GPU
>
> |                    | ZeRO-s1      | ZeRO-s2      | ZeRO-s3       |
> | ------------------ | ------------ | ------------ | ------------- |
> | Baseline (DP)      | $2\psi$      | $2\psi$      | $3\psi$       |
> | +GradAccum         | $2\psi$      | $N\psi+\psi$ | $3N\psi$ |
> | + GradAccum +AdamA | $N\psi+\psi$ | $N\psi+\psi$ | $3N\psi$ |
>
> Table2: Total memory footprint for each GPU
>
> |                    | ZeRO-s1 | ZeRO-s2 | ZeRO-s3 |
> | ------------------ | ------- | ------- | ------- |
> | Baseline (DP)      |    $\frac{1}{M}OS+G+W+A$     | $\frac{1}{M}OS+\frac{1}{M}G+W+A$ | $\frac{1}{M}OS+\frac{1}{M}G+\frac{1}{M}W+A$ |
> | + GradAccum        | $\frac{1}{M}OS+G+W+\frac{1}{N}A$ | $\frac{1}{M}OS+\frac{1}{M}G+W+\frac{1}{N}A$ | $\frac{1}{M}OS+\frac{1}{M}G+\frac{1}{M}W+\frac{1}{N}A$ |
> | + GradAccum +AdamA | $\frac{1}{M}OS+\frac{1}{L}G+W+\frac{1}{N}A$ | $\frac{1}{M}OS+\frac{1}{L}G+W+\frac{1}{N}A$ | $\frac{1}{M}OS+\frac{1}{L}G+\frac{1}{M}W+\frac{1}{N}A$ |
>
> From the tables above, we can jump to several conclusions:
>
> 1)   In all stages, as long as ZeRO is combined with GradAccum, the memory footprint will drop; The ZeRO-s1 communication volume remains unchanged, while the ZeRO-s2 and ZeRO-s3 communication volume is $\frac{N+1}{2}$ and $N$ times of the original ones;
>
> 2)   In all stages, as long as M is smaller than L, the memory footprint will decline when ZeRO+ GradAccum is combined with AdamA,; In addition, the fewer the GPUs, the greater the reduction compared with ZeRO+ GradAccum when combined with AdamA; In extreme cases (there is only one GPU), ZeRO-DP does not work at all, but we can still save almost all gradients and most activations;
>
> 3)   After we combine AdamA with both ZeRO-s2+ GradAccum and ZeRO-s3+ GradAccum, the communication volume will remain unchanged. On ZeRO-s1+ GradAccum, we will be $\frac{N+1}{2}$ times the original communication volume.

---

> > ### Author Response · Authors · 2022-11-08
> > **Response to Reviewer u5im (2)**
> >
> > As the throughput is affected by both computation time and communication time, we should analyze their respective proportion of total training time and tell the effect on throughput. Next, we will show the actual throughput effect when combining AdamA with ZeRO-s1/s3+ GradAccum, which leads to $\frac{N+1}{2}$/$\frac{N+2}{3}$ times of the original communication volume:
> >
> > 1)   In scenarios where computation time is absolutely dominant, the impact is small. For example, when training Bert-Large with 8 A100 GPUs using ZeRO-s1, we found that the computation time accounts for 98.3% of the total training time, and the communication time accounts for 1.7% of the total time. When we use AdamA with  ZeRO-s1+GradAccum (N takes the maximum 8 in the experiment), the total throughput will only drop about 5%, but in return, the memory occupied by activation and gradient will be greatly reduced. **We think this trade-off is very cost-effective when computation time is absolutely dominant**;
> >
> > 2)  In scenarios where communication time accounts for more, these model parameters are usually large, and memory saving in these scenarios has higher priority than training throughput. For example, when training the BERT-4B in Figure 6-b, each GPU needs at least 42.2GB when we applied ZeRO-s1. Even if ZeRO-s3 is used, each GPU needs at least 16.6GB of memory. For users who want to finetune BERT-4B with 8 pieces of 3080ti (12GB), they cannot run experiments. If we combine AdamA with ZeRO-s3+GradAccum, we only need 9.1GB memory for each GPU. In practice, when training BERT-4B with ZeRO-s3, the computation time accounts for 79.8%, and the communication time accounts for 20.2%. When we combine ZeRO-s3+GradAccum and AdamA (N=4), the throughput is about 38% slower. **We think that compared with the fact that we can finetune a larger model using GPUs with insufficient memory, this communication overhead is acceptable.**
> >
> > To sum up, we think that AdamA has many application scenarios, and the combination with ZeRO-DP can fill the gap it leaves in some scenarios.

---

> > ### Comment · Reviewer_u5im · 2022-11-09
> > **More questions about applicability**
> >
> > When looking at the applicability of this approach, I don't think it's sufficient to look at it independently across all configs.
> >
> > From my understanding, folks generally do *not* use gradient accumulation in combination with Zero-s2 or Zero-s3, precisely for the reasons in the chart you have. Increasing your communication volume by a factor of N is pretty bad, generally speaking.
> >
> > So, to summarize my takeaways:
> >
> > 1. Indeed, combining Zero-s1 + gradAccum with AdamA is not performant. We go from $2\psi$ to $N\psi + \psi$, which is a substantial increase in communication. **I would also note that this is one of the primary experiments in the paper, i.e. Fig 6b.**
> > 2. For Zero-s2/Zero-s3 + gradAccum + AdamA, although there's some reduction in memory usage, I'm not sure it's particularly significant (goes from $1/M$ to $1/L)$. Moreover, as noted in the chart, using any gradient accumulation with Zero-s2 or Zero-s3 results in a significant increase in communication.

---

> > > ### Author Response · Authors · 2022-11-10
> > > **Thank you for your response**
> > >
> > > We find that the discussion between you and us mainly focused on the overhead after the combination of AdamA and ZeRO-DP. However, strong compatibility is only a feature of AdamA, not our main contribution. I think we should reinterpret the insight that help we invent AdamA.
> > >
> > >
> > > 1) The current trend is that activation memory becomes extremely large (in many cases, the total memory footprint of activations is much more than the sum of weights, gradients and optimizer states), and the growing scale of models leads to large gradient memory. How ZeRO-DP works well depends on the number of GPUs. The more GPUs you have, the more memory reduction can be reached. However, the reality is that most researchers do not have so many GPUs.
> > >
> > >
> > > 2) We find that most researchers have very few GPUs (many researchers even have only one GPU), and they only finetune big models instead of pre-training when they get them. For such users, it is the highest priority to fit the model into the GPUs. However, ZeRO-DP even does not work for those users who only have one GPU. By contrast, our method can save equally considerable memory for each GPU, whether in single-GPU scenarios or multiple-GPU ones. Besides, AdamA alone can save more memory than ZeRO-DP in the very common scenarios where activation memory is dominant. In this way, our method makes up for some application scenarios that ZeRO-DP cannot cover or cannot perform well. I think this is the biggest motivation of this paper: we want to help those with limited devices.
> > >
> > >
> > > 3) Another contribution of our method is good compatibility. The motivation for us to talk about compatibility with other methods is that "after combining with other methods, the size of the models of your GPUs can train will be further improved". Our method compatibility gives users an additional choice. For those who have enough GPUs and can fit the model, they can indeed choose popular methods like ZeRO-DP. But for those who can't fit the model even after using ZeRO-DP, their highest priority is to fit it first. In these scenarios, I think the good compatibility of our method will help them.

---

> > > > ### Comment · Reviewer_u5im · 2022-11-11
> > > > **I think compatibility is an important characteristic of whether an optimization will be adopted.**
> > > >
> > > > I remain unconvinced that the applicability of AdamA is sufficient for me to recommend acceptance. Let's enumerate some of the cases we've referred to.
> > > >
> > > > Category 1: User has many GPUs
> > > >
> > > > 1. User has many GPUs, and is bound by activation memory. In this case, the best thing to do is to use approaches like gradient accumulation, or (regular) data-parallelism, or gradient checkpointing.
> > > > 2. User has many GPUs, and is now not bound by activation memory. In this case, you have the parameters ($\psi$), gradients ($\psi$), and Adam optimizer state ($2\psi$). In this case, the first thing we should use is Zero-S1 (instead of AdamA), as the optimizer state takes twice as much space as the gradients. Note that using Zero-S1 and AdamA are *mutually exclusive*, as we've established before that using them together substantially increases communication if we're using gradient accumulation. In addition, it is more valuable to use Zero-S1 here (compared to AdamA), as the optimizer state takes twice as much memory as the gradients.
> > > > 3. User has many GPUs, and is now bound by parameter and gradient memory. In this case, it depends on whether we're using gradient accumulation. If we aren't, then there is no difference between AdamA and Zero-S2. If we are, then our options all substantially increase communication bandwidth, and are thus undesirable. In practice, we would probably stop using gradient accumulation.
> > > >
> > > > Category 2:
> > > > 1. User has few GPUs (or a single GPU), and is bound by activation memory. In this case, it's similar to above, we'll probably use gradient accumulation/gradient checkpointing/w.e.
> > > > 2. User has few GPUs (or a single GPU), and is not bound by activation memory. As with above, I note that **gradient memory is only 25% of the total memory**. However, **I agree that AdamA has some application here**. But, as it's only a relatively small amount of the total memory, I'm unconvinced that it's that valuable. I would probably first target other memory saving approaches (like switching the optimizer from Adam).
> > > >
> > > > ### Conclusion
> > > > I agree that there are some cases where AdamA provides value. However, I think that those cases are not common enough/the benefits of AdamA are not large enough in those cases.
> > > >
> > > >
> > > > Additional note:
> > > >
> > > > > Besides, AdamA alone can save more memory than ZeRO-DP in the very common scenarios where activation memory is dominant.
> > > >
> > > > I don't think this is true. ZeRO-S1 + GradAccum will save more memory than AdamA + GradAccum.

---

> > > > > ### Author Response · Authors · 2022-11-16
> > > > > **Thank you for your further response**
> > > > >
> > > > > Thank you for your response! We think AdamA can help users explore the upper limit of trainable model size in few-GPU scenarios and has an absolute advantage in single-GPU scenarios. Reasons are as follows:
> > > > >
> > > > > 1. In large-scale model trainings, users are always bound by activation memory from the perspective of the entire system because the batch size is large. **The first thing we face is always to save activation memory instead of weights, gradients, and optimizer states.**
> > > > >
> > > > >       Data parallelism is the common method to amortize activation memory on each GPU, which depends on number of GPUs.  ZeRO-DP (a variant of data parallelism), focuses on many-GPUs scenarios. Thus, its memory optimization target turns to optimizer states, gradients and weights.
> > > > >
> > > > >       However, as AdamA focuses on few-GPU or single-GPU scenarios, it is still essential to take into account of activation and the other three parts together.
> > > > >
> > > > > 2. For few-GPU users, famous methods to reduce activation memory on each GPU include checkpointing and gradient accumulation. The checkpointing method will significantly slow down the training because we have measured the computation time during training and found it takes up the bulk. Therefore, gradient accumulation is our better choice to reduce activation memory.
> > > > >
> > > > > 3. **In few-GPU scenarios**, the memory can be bound by other memory components (like weights) after you reduce activation memory using gradient accumulation. If the GPU memory is still not enough for training, the only thing users can do is to try different methods one by one until the model can be fitted to the GPUs. We think that the reasonable sequence of trials is as follows (please note gradient accumulation has been used before we apply other methods). As the index increases, more memory reduction will be achieved, more negative impact will be put upon the training throughput. As we can see, AdamA can help users explore the upper limit of trainable model size in few-GPU scenarios
> > > > >
> > > > >     a) Gradient accum + ZeRO-S1: to reduce the memory of optimizer states.
> > > > >
> > > > >     b) Gradient accum + ZeRO-S2: to reduce the memory of optimizer states and gradients (1/M, M is the number of GPU, always to be 2 or 4 in few-GPU scenarios).
> > > > >
> > > > >     c) Gradient accum + AdamA + ZeRO-S2: to reduce the memory of optimizer states and more gradients (1/L, L is the number of layers in a model). The gradient memory footprint from 1/M to 1/L is huge, because M is always to be 2 or 3 in few-GPU scenarios, but L can be more than 100 or 1000. **For example, if we want to train a BERT-18.2B with 2 GPUs, the gradients memory on each GPU is 36.4GB when using ZeRO-S2. When using AdamA, the gradients memory is less than 1GB on each GPU.**
> > > > >
> > > > >     d) Gradient accum + AdamA + ZeRO-S3: to reduce the memory of optimizer states, weights, and gradients.
> > > > >
> > > > >     e) Other Off-loading methods.
> > > > >
> > > > > 4. **In single-GPU scenarios**, the memory is always hugely bound by activations. Previously, we used gradient accumulation to solve the problem. However, as AdamA + gradient accumulation reduces more memory than gradient accumulation alone and does the same with gradient accumulation in training throughput and model convergence, AdamA + gradient accumulation can be seen as an upgraded version of gradient accumulation, **Therefore, there is no reason to use gradient accumulation alone instead of AdamA + gradient accumulation now.**
> > > > >
> > > > > $\newline$
> > > > >
> > > > > Response to your note: In this sentence, I mean AdamA + GradAccum can save more memory than ZeRO-DP (without GradAccum ) where activation memory is dominant.

---

### Public Comment · ~Yuhan_Li3 · 2022-11-09
**Interesting paper**

The idea behind the paper sounds great. It's really helpful for researchers to save memory during training.

---

> ### Author Response · Authors · 2022-11-19
> **Thanks for your attention**
>
> We hope our paper can help lower the bar of training extremely large-scale models and benefit more researchers.

---

### Decision · Program_Chairs · 2023-01-20

**Decision:**

Reject

**Justification For Why Not Higher Score:**

Due to the following weakness, I would like to recommend a reject for this paper: 1) limited applicability, 2) drawback with more communication overhead, and 3) theoretical analysis should not be counted as a contribution.

**Justification For Why Not Lower Score:**

N/A

**Metareview: Summary, Strengths And Weaknesses:**

This paper provides a memory-efficient version of Adam which allows for gradient updates to be applied online during gradient accumulation. The modified version (AdamA) updates optimizer states directly, instead of accumulating the gradients after each step, so that the memory footprint of gradients can be reduced.

Strengths:
1. The modification makes sense in somehow and why the modification delivers performance improvement is understandable.
2. The author provides experimental results over vision and NLP tasks to demonstrate the effectiveness of the modification in some certain cases.
3. In general, idea is clearly presented. The paper is well written and easy to follow. The presentation is good.

Weaknesses:
1. Although the proposed method provides value in some scenarios, it has limited applicability in general cases in practice. The method mainly focuses on the case of few-GPU (or single-GPU) scenarios with trainable model size. Those cases are not common enough and the benefits are not large enough.
2. The method introduces more communication overhead. As we could have the scenarios where communication mainly affects the throughput, this is a drawback of the method.
3. The theory analysis provided by the paper for the convergence has flaws, which comes from the original Adam. This is not addressed by the authors.


**Summary Of Ac-Reviewer Meeting:**

N/A